# Public Perception towards Vaccines as Preventive Measures against a Twindemic (Seasonal Influenza and COVID-19): A Cross-Sectional Study from the Aseer Region, Saudi Arabia

**DOI:** 10.3390/vaccines11030558

**Published:** 2023-02-28

**Authors:** Sultan M. Alshahrani, Adel Alfatease, Khalid Orayj, Ali M. Alqahtani, Taha Alqahtani

**Affiliations:** 1Clinical Pharmacy Department, College of Pharmacy, King Khalid University, Abha 61441, Saudi Arabia; 2Pharmaceutics Department, College of Pharmacy, King Khalid University, Abha 62142, Saudi Arabia; 3Pharmacology Department, College of Pharmacy, King Khalid University, Abha 61441, Saudi Arabia

**Keywords:** influenza, COVID-19, vaccination, twindemic, Saudi Arabia

## Abstract

This study aimed to evaluate Saudi Arabian public perceptions toward influenza and COVID-19 immunization during the flu season. A cross-sectional self-administered, structured, and closed-questionnaire online survey was conducted on the general public. A total of 422 people willingly participated in the survey using several social media platforms from 15 May to 15 July 2021. Residents of Saudi Arabia aged 18 or older (eligible for COVID-19 vaccination) were included in the study and willing to answer questionnaires. The 422 participants who agreed to participate in the study completed the questionnaire. Thirty-seven percent of the participants were youth (18–25 years). More than 80% of the participants in the study agreed or strongly agreed that flu and COVID-19 vaccines must be mandatory for all populations. At the same time, 42.4% considered that the COVID-19 vaccine might positively impact the public and the economy in the future. Participants confirmed to have had COVID-19 or the flu since the beginning of the outbreak totaled 21.3%. Of the participants, 54% had sufficient knowledge about vaccine types and safety. Most of our participants (54.9%) agreed that preventive measures were still required, even with the existence of vaccines. Our study provides an overview of COVID-19′s influence on Saudi Arabia during the flu season. The Saudi Arabian government should consider preventive efforts to strengthen confidence in the health advantages offered by prospective immunization to prevent a twindemic of influenza and COVID-19.

## 1. Introduction

Pandemic RNA viruses are emerging rapidly, resulting in long-term human-to-human transmissions and clinical diseases. The world has faced a new coronavirus called Severe Acute Respiratory Syndrome 2 (SARS-CoV-2), an epidemiologic agent of COVID-19 (coronavirus disease 2019), which was first identified in Wuhan, China, in December 2019 and declared an international public health emergency within one month [1,2]. As of January 2023, the World Health Organization (WHO) reported about 660 million cases of COVID-19 and more than 6.7 million COVID-19-related deaths worldwide [3]. COVID-19 clinical symptoms vary, with the vast majority of infected people having only mild or no symptoms. At the same time, certain patients experience severe respiratory failure, requiring early and extended artificial ventilator assistance [4]. Patients with preexisting cerebrovascular diseases or strokes, liver disorders, hypertension, diabetes, COPD disease, and age over 60 are at an increased risk for COVID-19 infection and its mortality [5,6]. With the possibility of COVID-19 and influenza cocirculating in the fall/winter of 2020–2021, epidemiologists recommended high influenza vaccine coverage to help in reducing mortality and morbidity from influenza (and possibly from concurrent influenza diseases and COVID-19) while retaining the healthcare system’s ability to react to the pandemic [7,8].

The WHO reports that up to 650,000 deaths are linked annually to respiratory infections of seasonal influenza [9]. The Centers for Disease Control and Prevention (CDC) projected 410,000–740,000 influenza-related hospitalizations and 24,000–62,000 influenza-related deaths in the United States from 1 October 2019 to 4 April 2020. Influenza and COVID-19 might be symptomatically identical and lead to more severe-course coinfections, consequences, or death outcomes. Moreover, COVID-19 spreads among high-risk categories the same way as seasonal influenza, which can harm older people and those with chronic comorbidity, including obese people and long-term care patients [10,11,12,13]. In a Brazilian study assessed by more than 92,000 COVID-19 patients, the death risks were 17% lower, the intensive care requirements were 8%, and the invasive response odds were 18% for those who obtained an influenza vaccine [14]. In another study, COVID-19-related deaths in nursing homes in Europe were between 40–80% [13]. Moreover, vaccination intake during the global H1N1 pandemic in 2009 was relatively low among the general public, with several countries having less than 50% of the projected coverage. However, people who were vaccinated with the influenza vaccine in the preceding season were more likely to avoid influenza during the pandemic [15]. Thus, vaccination is crucial to lowering the risk of influenza infection and coinfection in high-risk populations with SARS-CoV-2 problems, diagnosis issues, and inadequate management [16]. According to a study in Saudi Arabia, 199 (55%) participants received the last seasonal influenza vaccine (2019–2020), with just 56 (15.4%) receiving the seasonal influenza vaccine annually, whereas 68 (18.7%), in the preceding five years, had not received the seasonal influenza vaccine. Among these participants, 55.9% are willing to receive the next COVID-19 vaccination [17]. In Saudi Arabia, the designing and development of a proper theoretical framework, as well as monitoring vaccine implementation and their outcomes by the healthcare authorities, should help public health specialists and healthcare providers in designing strategies that increase vaccination coverage and improve general acceptance of vaccines especially during the flu seasons.

Considering the impending “twindemic” of influenza and COVID-19, comprehensive education campaigns, structural reforms, and novel influenza vaccination approaches must encourage the people of Saudi Arabia to immunize, ultimately increasing their trust in the health benefits of potential SARS-CoV-2 vaccines when available to the general public. Our goal was to assess the public’s perception toward influenza and COVID-19 immunization during the flu season and how it would improve during similar circumstances in the future.

## 2. Materials and Methods

### 2.1. Study Design and Sample Size Calculation

This cross-sectional study was conducted between May and July 2021 among the public in the Aseer region, Saudi Arabia. It was allocated through several social media platforms (Facebook, Twitter, and Whatsapp) for participants. Data collection continued for three months (15 May to 15 July). The population of the Aseer region (the southern region of Saudi Arabia) is about 2.2 million. The α-level was set to 5%, and the confidence interval was set to 95% with 5% precision. The sample size (n) was calculated based on this formula to estimate the number of subjects or patients required for this study:n = (Z^2^ p(1 − p))/d^2^

where n = sample size, Z = Z statistic for the confidence level (the literature has often selected Z = 1.96 to provide a good estimation of sample size), and p = expected prevalence or proportion. Using this equation, the number of subjects that should be included became 386. We added 10% to avoid any errors in completing the survey. The total number of participants who volunteered for the survey was 422 (response rate 84.4% out of 500 surveys distributed).

### 2.2. Questionnaire

Utilizing previously published studies [16,18], a self-administered, structured, and closed questionnaire was designed and subsequently modified to suit the general public of the Aseer region, Saudi Arabia. Four researchers revised the survey scientifically, and a linguistic professional evaluated the study’s language. The questionnaire was prepared in English and translated into Arabic by a specialized translator. To make the survey more reliable, inclusion criteria were made. Residents of the Aseer region aged 18 or older (eligible for COVID-19 vaccination) were included in the study and willing to answer questionnaires. Participants that were unwilling to provide their consent were omitted. The data on public attitudes toward vaccines for COVID-19 and the flu during the flu season were collected. The survey included two parts: part 1 focused on demographics, and part 2 on participants’ perspectives. The participants’ part (knowledge and attitudes) was evaluated on a five-points Likert’s scale, including the options: strongly agree, agree, neutral, disagree, and strongly disagree. To check whether the expression of items could be understood, a pilot test was conducted among eleven participants. The Cronbach alpha factor was calculated for the pilot study and assessed as 0.77.

The questionnaire was written in Google Forms, which required 5 min on average to be completed. The survey was distributed via a QR code during the vaccination campaign at two main vaccination centers in the Aseer region.

### 2.3. Statistical Analysis

We evaluated the surveys and cleaned, coded, and entered the data into SPSS version 20 for statistical analysis (IBM Corp., Armonk, NY, USA). We used both inferential and descriptive statistics to achieve our results. The issues presented in the questionnaire were either positive or negative for COVID-19 vaccinations. Strongly agree and agree were regarded as positive attitudes, whereas neutral, disagree, and strongly disagree were considered negative attitudes (control). Logistic regression was then adapted to discover independent factors with a favorable social media attitude. Independent factors included sex, age, university level, monthly income, employment status, and residence area, whereas dependent factors were sex, age, university level, monthly income, employment status, and residential area. They were previously treated as factors and included in the logistical regression model. In order to detect the linkage between dependent and independent variables, odds ratios (ORs) and 95% confidence intervals (CIs) were used. *p* was considered significant at <0.05.

### 2.4. Ethical Considerations

The data collection processes were standardized, and no personal information about the participants was collected or stored. The remaining information was kept confidential during the study and data analysis. The participants were asked for their consent before the beginning of the survey. They were not asked for their ID or any personal information. Participation in the study was completely voluntary.

We followed the World Medical Association [WMA] Declaration of Helsinki: Ethical standards for medical research involving human people, as amended by the 59th WMA (ECM#2021-5415), Seoul, Korea, in this study. No personally identifiable information about the participants was collected. In addition, the Research Ethics Committee at King Khalid University (HAPO-06-B-001) reviewed and agreed on this project: Approval No. ECM#2021-5414; Approval date 2 May 2020.

## 3. Results

### 3.1. Demographic Characteristics

According to our results, a total of 422 participants aged from 18 to >55 years old completed the questionnaire. Most of the participants were youth (18–25 years, 37.2%), followed by those aged 26–35 years (35.5%), 36–45 years (21.5%), 46–55 years (4%), and >55 (1.6%). There were 311 female participants (73.6%) and 111 male participants (26.3%). Individuals with a university degree accounted for 72.7% of those who graduated, with 19.1% having a secondary education or less and 8% having a postgraduate degree. Among those who participated, 56.8% claimed to have a monthly salary of <5000 SR, whereas 24.8% reported having a monthly income between 8000 and 15,000 SR (Table 1). A large percentage of these participants are urbanites (82.9%). However, 21.3% of participants stated that they had been diagnosed with COVID-19 or the flu since the beginning of the COVID-19 pandemic. Furthermore, 21.5% had received flu vaccinations since the beginning of the COVID-19 outbreak.

Table 2 summarizes the participants’ perceptions of influenza and COVID-19 immunization during the flu season in Saudi Arabia, as expressed through social media. Of the participants, 48.8% have strongly agreed that flu and COVID-19 vaccines must be mandatory for all populations, whereas 42.4% considered that the COVID-19 vaccine might positively impact the public and economy in the future. Furthermore, 54% strongly agreed that they would receive flu shots this year, and 40.7% and 50.7% advised others to take flu and COVID-19 vaccines, respectively. Fifty-four percent of the participants have sufficient knowledge about COVID-19 vaccine types and safety. Additionally, the participants agreed (44.5%) or strongly agreed (34.5%) that COVID-19 cases might increase during the flu season, negatively impacting the healthcare system. Our participants strongly agreed (54.9%) that preventive measures are still necessary, even in the presence of vaccines.

### 3.2. A Logistic Regression Model Examining Factors Affecting Sufficient Knowledge of Flu and COVID-19 Vaccines

Table 3 shows the logistic regression model results, which revealed a significant association between some independent variables and positive attitudes toward social media. Male participants were less likely to be influenced by social media when deciding to receive COVID-19 vaccines (*p* = 0.041, OR: 0.679, CI: 0.468–0.985). Compared to participants with lower education levels, participants with a university education level or with a postgraduate degree level were less likely to share information obtained from social media about the vaccine without ensuring that the information was correct (*p* = 0.01, OR: 0.546, CI: 0.344–0.866 and *p* = 0.003, OR: 0.331, CI: 0.158–0.693, respectively). Additionally, postgraduate-level participants were less likely to have a positive attitude toward social media as a source of information regarding COVID-19 vaccines. The data are depicted in Table 4.

## 4. Discussion

We designed this study to evaluate the public’s perception toward flu vaccines and to assess the health attitudes and behaviors associated with ongoing flu season experiences with COVID-19. The study was a cross-sectional study using survey data with a sample size from one region of Saudi Arabia; therefore, only limited generalizability of these findings can be claimed. However, to our knowledge, this is one of the few studies to have evaluated Saudi Arabia’s attitude and concern toward receiving COVID-19 vaccination during seasonal influenza. Out of 422 participants, 21.3% have been diagnosed with COVID-19 or the flu since the beginning of the COVID-19 pandemic, and 21.5% have received flu vaccines in that time. According to a recent study conducted in Saudi Arabia, 61.2% of health workers were willing to take the COVID-19 vaccine, and 55.9% received the seasonal influenza vaccine in previous years [17]. The strongest predictive factor for vaccine acceptance was that most participants agreed (39.5%) or strongly agreed (48.8%) that flu and COVID-19 vaccines must be mandatory for all populations. According to a few prior studies, social and economic factors may influence receiving COVID-19 vaccines. However, according to a previous British study, ethnic group healthcare professionals, particularly younger females, were inversely associated with COVID-19 vaccination uptake, similar to prior pandemics [19,20,21]. In our study, most participants demonstrated positive attitudes and sufficient knowledge regarding influenza and COVID-19 immunization during flu season. Most of our participants through social media were women aged 18–45. Vaccine intention is connected to the prior acceptance of specific vaccine types. Previous studies have identified behavior as a significant predictor of future vaccine practice. By contrast, a low acceptance rate for the influenza vaccine was observed among healthcare professionals from Saudi Arabia. Similar outcomes were reported in the USA, Turkey, and Australia for accepting COVID-19 and influenza vaccinations [22,23,24,25,26,27]. Our research found statistical significance between age groups (36–45 years and >55 years), education levels, and employment in logistic regression models, possibly due to younger age groups (flu and COVID-19 vaccination).

The vaccine may be the most efficient way of preventing COVID-19, since it saves lives and reduces the loss of productivity since prepandemic activities have resumed [28]. Mass COVID-19 vaccination strategies may reduce asymptomatic and symptomatic COVID-19 symptoms with a coverage of 75–81% for AstraZeneca and 70% for the Pfizer and Moderna vaccines. Regarding the estimated efficacies for reduced hospitalization, Moderna and Pfizer have 80%, and AstraZeneca has 78%. Mass vaccination has demonstrated outstanding efficacy in decreasing the health requirements of nations facing an outbreak of COVID-19 [28,29]. Many participants reported similar outcomes and considered the COVID-19 vaccine as positively impacting the public and economy. Most participants agreed (44.3%) or strongly agreed (42.5%) that vaccines must positively impact public attitudes. The Pfizer and Moderna mRNA vaccines have shown efficacies of 94–95% in preventing symptomatic COVID-19 [30,31]. Our study found statistical significance between age groups (26–35 years and >55 years), education levels (postgraduates), employment (private sector jobs), and receiving flu vaccines in logistic regression models regarding COVID-19 vaccine safety. The Minister of Health highlighted the safety of various COVID-19 vaccines in two-dose vaccinations [32]. There are about 34 million residents in Saudi Arabia. Vaccination coverage of approximately 67% is required to achieve herd immunity from COVID-19 infection [33,34]. Similarly, our participants agreed (24.4%) or strongly agreed (43.3%) that vaccines improve human immunity. According to previous studies, the Saudi Arabian people are hesitant to receive vaccines, as there are several reports of side effects [35,36]. However, the immunization rates among the elderly in Saudi Arabia are high. The Ministry of Health of Saudi Arabia reported that 98% of vaccinated persons in Hafar Al-Batin city, followed by Al-Ahsa (93%), Al-Qurayyat (93%), Bisha (86%), Riyadh (83%), Eastern province (80%), and Taif (80%), have received at least one vaccine dose [37]. We found that approximately 50.7% of participants advised others to receive COVID-19 vaccines and have sufficient knowledge of them.

Various studies show that the positivity of the seasonal flu vaccine in public opinion strongly influences COVID-19 vaccine acceptance [38,39]. Our participants reported positive knowledge about the impact of COVID-19 vaccines on seasonal influenza, whereas most participants agreed (41.9%) or strongly agreed (45.9%) that increasing COVID-19 cases during flu season has negatively impacted the healthcare system. We found that 21.5% of our study participants received the seasonal influenza vaccine since the beginning of COVID-19. Moreover, half of our participants strongly agreed that they would receive flu shots this year during the pandemic. According to a previous study, 55% of Saudi Arabian participants received seasonal influenza vaccines in the 2019–2020 season [17].

People are currently facing an influenza season alongside a worldwide pandemic, causing a huge burden on healthcare systems and increasing the risk of COVID-19 and influenza coinfection [40]. Most participants in the study also considered that COVID-19 cases might increase during the flu season. Since the influenza vaccine effectively protects against influenza and COVID-19, vaccine-hesitant participants may feel more encouraged [41]. Our participants strongly recommend that preventive measures should still be required even in the presence of vaccines.

The study was among the first studies conducted in Saudi Arabia to reveal the population prespective toward twindemic vaccination against flu and COVID-19 during the drastic conditions of the pandemic. However, there are some limitations to this study, as it utilized self-reported survey from one region of the country, since the community-based data collection was not feasible at the time of pandemic and curfew. In addition, the endogenous variables such as psychological states during the pandemic and limited information about the new vaccines may have played a significant role in participants’ attitudes. Further research is needed to cover all parts of the country and to be conducted on regular times where endogeneity variables are eliminated.

## 5. Conclusions

In conclusion, most participants were inclined to accept the COVID-19 vaccine while considering the influence of COVID-19 on the future attitudes of the public and the economy. Furthermore, despite having received the flu vaccination in previous years, the participants claim that COVID-19 negatively impacts flu medications in hospitals and pharmacies. Another notable point is that even if the COVID-19 and seasonal influenza vaccines are available during the pandemic, preventive measures are still required. Healthcare authorities in Saudi Arabia are required to provide more encouraging efforts and awareness campaigns toward the uptake of public flu vaccination.

## Figures and Tables

**Table 1 vaccines-11-00558-t001:** Demographic information about the participants.

Demographic	n (%) N = 422
Sex	
Male	111 (26.3)
Female	311 (73.7)
Age group	
18–25 years	157 (37.2)
26–35 years	150 (35.5)
36–45 years	91 (21.6)
46–55 years	17 (4)
>55 years	7 (1.7)
Educational level	
High school or lower	81 (19.2)
University	307 (72.7)
Postgraduate degree	34 (8.1)
Employment status	
Unemployed	165 (39.1)
Student	108 (25.6)
Government sector job	108 (25.6)
Private sector job	41 (9.7)
Monthly income (Median =9500 SR)	
<5000 Saud Riyal (SR)	240 (56.9)
5000–8000 SR	56 (13.3)
8000–15,000 SR	105 (24.9)
>15,000 SR	21 (4.9)
Residence area	
Rural	72 (17.1)
Urban	350 (82.9)
Have you received flu vaccines since the beginning of the COVID-19 pandemic? (yes)	91 (21.5)

**Table 2 vaccines-11-00558-t002:** Participants’ viewpoints towards the questionnaire.

	N (%)	N (%)	N (%)	N (%)	N (%)
Question	Strongly Disagree	Disagree	Neutral	Agree	Strongly Agree
In the last decade, vaccinations like chickenpox vaccines saved the public from disasters, so I think the flu and COVID-19 vaccines should be mandatory for all populations.	3 (0.7)	2 (0.4)	44 (10.4)	**167 (39.5)**	**206 (48.8)**
Taking the COVID-19 vaccine will positively impact the public and economy in the future.	2 (0.4)	1 (0.2)	53 (12.5)	**187 (44.3)**	**179 (42.4)**
You will take the flu shot this year.	6 (1.4)	21 (4.9)	69 (16.3)	**98 (23.2)**	**228 (54)**
All types of COVID-19 vaccines are safe in general.	2 (0.4)	3 (0.7)	91 (21.5)	**108 (25.5)**	**218 (51.6)**
Vaccines are important to improve human immunity.	1 (0.2)	2 (0.4)	133 (31.5)	**103 (24.4)**	**183 (43.3)**
I advise others to take the flu shot.	1 (0.2)	4 (0.9)	151 (35.7)	**94 (22.2)**	**172 (40.7)**
I advise others to take the COVID-19 vaccine.	0 (0)	1 (0.2)	138 (32.7)	**69 (16.3)**	**214 (50.7)**
Your knowledge regarding COVID-19 vaccine types is good.	3 (0.7)	12 (2.8)	72 (17)	**104 (24.6)**	**231 (54.7)**
COVID-19 has a negative impact on flu medications in hospitals and pharmacies.	11 (2.6)	11 (2.6)	73 (17.2)	**185 (43.8)**	**142 (33.6)**
COVID-19 cases are supposed to increase during flu season.	9 (2.1)	6 (1.4)	73 (17.2)	**188 (44.5)**	**146 (34.5)**
Increasing COVID-19 cases during flu season will negatively impact the healthcare system.	3 (0.7)	3 (0.7)	45 (10.6)	**177 (41.9)**	**194 (45.9)**
Following preventive measures is still important even in the presence of vaccines.	1 (0.2)	2 (0.4)	32 (7.5)	**155 (36.7)**	**232 (54.9)**

Bolded numbers indicate the most common answers.

**Table 3 vaccines-11-00558-t003:** Logistic regression model examining factors affecting the sufficient knowledge towards flu and COVID-19 vaccines.

	Dependent Factors	In the Last Decade, Vaccinations like Chickenpox Vaccines Have Saved the Public from Disasters, So I Think Flu and COVID-19 Vaccines Should Be Mandatory for All Populations.	Taking COVID-19 Vaccine Will Positively Impact the Public and Economy in the Future	You Will Take the Flu Shot This Year	All Types of COVID-19 Vaccines Are Safe in General	Vaccines Are Important to Improving Human Immunity	I Advise Others to Take the Flu Shot
Independent Factors							
		OR (95% CI) *p*-Value	OR (95% CI), *p*-Value	OR (95% CI) *p*-Value	OR (95% CI) *p*-Value	OR (95% CI) *p*-Value	OR (95% CI) *p*-Value
**Sex (ref = female)**		1.252 (0.558–2.812) 0.585	0.688 (0.312–1.517) 0.354	1.562 (0.771–3.163) 0.216	1.078 (0.573–2.025) 0.816	1.496 (0.837–2.675) 0.174	1.35 (0.767–2.373) 0.298
**Age**							
18–25		ref	ref	ref	ref	ref	ref
26–35 years		2.111 (0.73–6.1) 0.168	**3.76 (1.341–10.54) 0.012**	1.148 (0.522–2.524) 0.731	**2.259 (1.038–4.918) 0.04**	0.887 (0.456–1.728) 0.725	0.843 (0.433–1.64) 0.615
36–45 years		**3.407 (1.005–11.547) 0.049**	3.07 (0.964–9.77) 0.058	2.095 (0.813–5.4) 0.126	2.323 (0.938–5.751) 0.068	0.895 (0.414–1.934) 0.778	0.733 (0.342–1.572) 0.424
46–55 years		7.119 (0.701–72.325) 0.097	1.093 (0.245–4.874) 0.907	0.824 (0.193–3.513) 0.794	1.077 (0.269–4.309) 0.917	1.57 (0.407–6.062) 0.513	0.95 (0.27–3.339) 0.936
>55 years		**0.067 (0.009–0.506) 0.009**	**0.052 (0.006–0.432) 0.006**	**0.008 (0–0.144) 0.001**	**0.077 (0.011–0.556) 0.011**	**0.13 (0.02–0.823) 0.03**	**0.117 (0.018–0.78) 0.027**
**Education level**							
High school or lower		ref	ref	ref	ref	ref	ref
University		**3.288 (1.402–7.714) 0.006**	**2.428 (1.071–5.503) 0.034**	1.798 (0.919–3.516) 0.087	1.05 (0.501–2.201) 0.897	1.567 (0.88–2.791) 0.127	1.767 (0.992–3.145) 0.053
Postgraduate degree		2.911 (0.738–11.485) 0.127	1.048 (0.256–4.285) 0.948	2.197 (0.635–7.602) 0.214	**0.21 (0.066–0.669) 0.008**	1.403 (0.504–3.904) 0.517	1.862 (0.67–5.176) 0.233
**Monthly income (Median = 9500 SR)**							
Less than 5000 SR		ref	ref	ref	ref	ref	ref
5000–8000 SR		2.5 (0.644–9.703) 0.186	1.159 (0.441–3.05) 0.764	**3.679 (1.302–10.393) 0.014**	1.156 (0.526–2.541) 0.719	1.712 (0.821–3.572) 0.152	2.051 (0.995–4.228) 0.051
8000–15,000 SR		1.249 (0.346–4.507) 0.734	3.795 (0.976–14.749) 0.054	0.57 (0.231–1.403) 0.221	2.411 (0.877–6.627) 0.088	1.051 (0.483–2.288) 0.9	1.039 (0.484–2.228) 0.923
>15,000 SR		0.678 (0.134–3.427) 0.638	4.659 (0.505–43.017) 0.175	0.298 (0.081–1.088) 0.067	1.114 (0.303–4.088) 0.871	0.914 (0.295–2.83) 0.876	0.56 (0.186–1.685) 0.302
**Employment status**							
Unemployed		ref	ref	ref	ref	ref	ref
Student		0.464 (0.149–1.448) 0.186	0.813 (0.285–2.319) 0.699	0.55 (0.234–1.293) 0.171	0.851 (0.375–1.93) 0.699	0.641 (0.311–1.319) 0.227	**0.47 (0.228–0.967) 0.04**
Government sector job		**0.101 (0.026–0.401) 0.001**	**0.172 (0.043–0.684) 0.012**	0.512 (0.186–1.41) 0.195	0.38 (0.133–1.084) 0.071	0.489 (0.211–1.137) 0.096	0.464 (0.202–1.067) 0.071
Private sector job		**0.239 (0.059–0.96) 0.044**	**0.231 (0.07–0.764) 0.016**	0.372 (0.125–1.108) 0.076	**0.22 (0.086–0.565) 0.002**	1.027 (0.398–2.649) 0.956	0.851 (0.339–2.137) 0.732
**Residence area (ref = rural)**		0.774 (0.301–1.994) 0.596	**0.155 (0.039–0.614) 0.008**	**0.339 (0.143–0.807) 0.015**	0.42 (0.189–0.932) 0.033	0.942 (0.523–1.696) 0.842	0.677 (0.374–1.228) 0.2
**Have you been diagnosed with COVID-19 or Flu since the beginning of the COVID-19 pandemic? (ref = no)**		1.326 (0.576–3.055) 0.508	1.036 (0.473–2.272) 0.929	0.969 (0.509–1.845) 0.924	0.665 (0.369–1.2) 0.175	1.564 (0.887–2.758) 0.122	1.224 (0.714–2.098) 0.462
**Have you received flu vaccines since the beginning of the COVID-19 pandemic? (ref = no)**		1.983 (0.799–4.925) 0.14	**3.223 (1.218–8.529) 0.018**	**46.977 (6.286–351.085) 0.001**	**2.189 (1.105–4.335) 0.025**	**2.753 (1.468–5.162) 0.002**	**3.958 (2.101–7.457) 0.001**

Bolded numbers indicate significant *p* values.

**Table 4 vaccines-11-00558-t004:** A logistic regression model examining factors affecting sufficient knowledge towards flu and COVID-19 vaccines.

	Dependent Factors	I Advise Others to take COVID-19 Vaccine	Your Knowledge Regarding COVID-19 Vaccines Types is Good	COVID-19 Has A negative Impact on Flu Medications in Hospitals and Pharmacies	COVID-19 Cases Are Supposed to Increase during Flu Season	Increasing COVID-19 Cases during Flu Season Will Negatively Impact the Healthcare System	Following Preventive Measures Are Still Important even in the Presence of Vaccines
Independent Factors							
		OR (95% CI) *p*-Value	OR (95% CI), *p*-Value	OR (95% CI) *p*-Value	OR (95% CI) *p*-Value	OR (95% CI) *p*-Value	OR (95% CI) *p*-Value
**Sex (ref = female)**		1.009 (0.575–1.773) 0.974	0.858 (0.441–1.67) 0.652	1.011 (0.528–1.936) 0.974	1.18 (0.609–2.286) 0.624	0.569 (0.249–1.297) 0.18	0.604 (0.207–1.765) 0.357
**Age**							
18–25		ref	ref	ref	ref	ref	ref
26–35 years		1.067 (0.551–2.066) 0.848	**5.378 (2.397–12.067) 0.001**	1.104 (0.529–2.301) 0.793	**2.229 (1.019–4.872) 0.045**	**4.678 (1.789–12.235) 0.002**	**6.539 (1.99–21.488) 0.002**
36–45 years		0.949 (0.442–2.036) 0.893	**4.662 (1.901–11.43) 0.001**	1.3 (0.546–3.095) 0.554	1.93 (0.799–4.664) 0.144	**4.184 (1.355–12.922) 0.013**	**19.856 (3.241–121.655) 0.001**
46–55 years		1.294 (0.36–4.647) 0.693	1.702 (0.476–6.085) 0.413	0.803 (0.225–2.861) 0.735	1.445 (0.363–5.752) 0.601	5.493 (0.905–33.341) 0.064	3.616 (0.568–23.012) 0.173
>55 years		**0.151 (0.024–0.953) 0.044**	0.179 (0.026–1.216) 0.078	0.402 (0.069–2.33) 0.309	0.334 (0.055–2.016) 0.232	0.303 (0.041–2.227) 0.241	0.057 (0.005–0.623) 0.019
**Education level**							
High school or lower		ref	ref	ref	ref	ref	ref
University		1.695 (0.955–3.009) 0.072	1.155 (0.571–2.335) 0.689	**1.98 (1.076–3.642) 0.028**	1.324 (0.663–2.645) 0.427	**2.337 (1.099–4.97) 0.028**	1.47 (0.574–3.764) 0.422
Postgraduate degree		2.065 (0.729–5.846) 0.172	1.577 (0.467–5.324) 0.463	1.874 (0.596–5.892) 0.282	0.37 (0.123–1.11) 0.076	5.613 (0.605–52.063) 0.129	2.533 (0.229–28.009) 0.449
**Monthly income**							
Less than 5000 SR		ref	ref	ref	ref	ref	ref
5000–8000 SR		1.906 (0.913–3.976) 0.086	1.283 (0.537–3.066) 0.575	**2.775 (1.097–7.019) 0.031**	0.781 (0.361–1.69) 0.53	3.44 (0.76–15.567) 0.109	3.963 (0.49–32.061) 0.197
8000–15,000 SR		1.473 (0.668–3.252) 0.337	0.815 (0.323–2.053) 0.664	1.741 (0.682–4.443) 0.246	0.935 (0.372–2.348) 0.886	0.686 (0.216–2.182) 0.523	0.358 (0.094–1.372) 0.134
>15,000 SR		0.905 (0.3–2.735) 0.86	0.375 (0.108–1.303) 0.123	2.828 (0.638–12.53) 0.171	0.436 (0.129–1.474) 0.182	Cannot be computed	Cannot be computed
**Employment status**							
Unemployed		ref	ref	ref	ref	ref	ref
Student		0.724 (0.354–1.482) 0.377	**2.512 (1.075–5.867) 0.033**	1.203 (0.529–2.735) 0.659	1.044 (0.458–2.381) 0.918	2.179 (0.821–5.78) 0.118	**3.974 (1.077–14.668) 0.038**
Government sector job		0.436 (0.186–1.019) 0.055	0.434 (0.161–1.174) 0.1	0.535 (0.195–1.464) 0.223	0.822 (0.303–2.225) 0.699	0.849 (0.228–3.156) 0.806	0.872 (0.183–4.153) 0.863
Private sector job		0.899 (0.356–2.266) 0.821	0.516 (0.191–1.393) 0.192	1.073 (0.39–2.953) 0.891	0.784 (0.296–2.074) 0.624	2.16 (0.419–11.122) 0.357	1.42 (0.247–8.152) 0.694
**Residence area (ref = rural)**		0.839 (0.466–1.51) 0.558	**0.363 (0.154–0.859) 0.021**	0.502 (0.244–1.031) 0.06	0.454 (0.206–1.001) 0.05	0.512 (0.203–1.29) 0.156	**0.168 (0.037–0.771) 0.022**
**Have you been diagnosed with COVID-19 or Flu since the beginning of the COVID-19 pandemic? (ref = no)**		1.21 (0.702–2.086) 0.493	1.26 (0.66–2.404) 0.484	**2.293 (1.15–4.572) 0.018**	1.902 (0.964–3.751) 0.064	2.409 (0.937–6.197) 0.068	**14.373 (1.708–120.975) 0.014**
**Have you received flu vaccines since the beginning of the COVID-19 pandemic? (ref = no)**		**3.092 (1.649–5.795) 0.001**	1.709 (0.869–3.365) 0.121	0.891 (0.488–1.624) 0.705	1.031 (0.548–1.939) 0.925	1.502 (0.608–3.715) 0.378	**5.654 (1.155–27.677) 0.033**

Bolded numbers indicate significant *p*-value.

## Data Availability

The data of this study are available upon reasonable request from the authors.

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
