# Peer review of "Public Perception towards Vaccines as Preventive Measures against a Twindemic (Seasonal Influenza and COVID-19): A Cross-Sectional Study from the Aseer Region, Saudi Arabia"

_vaccines, 2023, doi:10.3390/vaccines11030558_

Round 1

Reviewer 1 Report (Previous Reviewer 3)

The manuscript is greatly improved.  However, there are still some editing issues that the authors should consider and address.  The following are suggestions/comments regarding those issues.  Line 14, "...immunization during the flu season."  Line 44, "age>60 are at an increased risk for COVID-19 ...".  Lines 76-78, "immunization during the flu season and how would that improve during similar circumstances in the future."  Line 98, "Utilizing previously published ...".  Line 103, "... pilot study participants and assessed as 0.77."  Line 106, "answer the questionnaires."  Table 3, first column, "In the last decade, vaccinations like chickenpox vaccines have saved the public ...".  Line 253, "during the flu season has negatively impacted the healthcare system."  Line 261, "... might increase during the flu season."  Line 270, "... psychological state during the pandemic ...".  Line 272, "Further research is needed to cover all ...".  Line 279, "... negatively impacts the flu medications in ...".

Author Response

Reviewer 2 Report (New Reviewer)

This is an interesting paper on a timely and important subject in preventive attitudes, knowledge, and practice from a cross-sectional study design.  The findings have commonly confirmed the importance of knowledge and attitudinal effect on vaccinations. The paper could be improved in the following areas:

1. Lengthy statements could be shortened.

2. A theoretical framework on preventive care could be introduced to guide the analysis/

3. Preventive knowledge is a latent construct measured by multiple items in the Likert Scale.  To simplify the table presentation, it is suggested that a composite score could be constructed. Thus, the validity and reliability of the measurement scale could be presented.

4. Limitations of the study should be noted in the discussion section.

5. Health policy recommendations relevant to preventive actions should be presented and elaborated on in the concluding section.

Round 2

Reviewer 2 Report (New Reviewer)

Please correct a typo in Line 281:  indogenous to endogenous.  Authors did not discuss the comment on validity and reliability issues associated with the knowledge or attitude scales.  Otherwise, the paper is acceptable.

Author Response

We would like to thank the reviewer for the valuable comments.

1- The typo of endogenous was corrected.

2- The methodology (questionnaire section) was revised and edited as per the reviewer's comments as:

"

2.2. Questionnaire

Utilizing previously published studies [16,18], a self-administered, structured, and closed questionnaire was designed and subsequently modified to suit the general public of the Aseer region, Saudi Arabia. Four researchers revised the survey scientifically, and a linguistic professional evaluated the study’s language. The questionnaire was prepared in English and translated into Arabic by a specialized translator. To make the survey more reliable, inclusion criteria were made. Residents of the Aseer region aged 18 or older (eligible for COVID-19 vaccination) were included in the study and willing to answer questionnaires. Participants that were unwilling to provide their consent were omitted. The data on public attitudes toward vaccines with COVID-19 and flu during the flu season were collected. The survey included two parts: part 1 focused on demographics, and part 2 on participants’ perspectives. The participants’ part (knowledge and attitudes) was made on 5-points Likert’s scale as strongly agree, agree, neutral, disagree, strongly disagree. In order to check whether the expression of items could be understood, a pilot test was conducted among eleven participants. The Cronbach alpha factor was calculated for the pilot study and assessed as 0.77.

The questionnaire was written in Google format, which required 5 minutes on average to be completed. The survey was distributed via a QR code during the vaccination campaign at two main vaccination centers in the Aseer region."

This manuscript is a resubmission of an earlier submission. The following is a list of the peer review reports and author responses from that submission.

Round 1

Reviewer 1 Report

The authors have evaluated the Saudi Arabian public perceptions toward influenza and COVID-19 immunization during flu season among a group of 422 individuals during May and July 2021. The title and abstract cover the main aspect of the work. However, the abstract content could be improved by avoiding repeating ideas, enhancing the most scientific results of the study, and improving the English language. The introduction briefly presents the impact of SARSCOV2 and influenza worldwide but could be improved by adding new updated information about how the COVID-19 pandemic evolved until today. The study aims to assess public perceptions toward influenza and COVID-19 immunization during flu season, but I would suggest the authors update these aims by underlying why or how the study results could impact health today. 

The applied questionnaire was not available as supplementary material.

In the results section, the writing should be persistent in style, either % or percent. 

In Table 1, Employment status, "privet" is not in English.

Line 272, "patients41" what does mean?

Reviewer 2 Report

The article “Public Perception towards Vaccines as Preventive Measures against Twindemic (Seasonal Influenza and COVID-19): A Cross-Sectional Study from Aseer Region, Saudi Arabia” investigates data obtained from a questionnaire. These types of studies may have some merit if conducted properly. However, in this case there are several serious problems with the design, so I can’t recommend this paper to publish in Vaccines.

The main issue is with sample collection, which is biased and not suitable to represent the population.

              The survey was collected in vaccination centers, among people more inclined to vaccinate themselves.

              The distribution was via QR code, which needs a certain amount of technical knowledge, therefore older people did not fill the survey (proved by the results).

Authors should mention median salary levels of the country, else the salary numbers mean little to foreign readers.

The determination of sufficient knowledge is highly questionable.

Reviewer 3 Report

A very interesting, informative and well written manuscript.  However, there are some editing issues that the authors should address.  The following are suggestions/comments regarding the editing issues.  Line14, "...immunization during the flu season."  Line 32, "...rapidly, which results in long-term ...".  Line 36, "in Wuhan, China in December ...".  Line 37, "...July 2021, the World Health Organization (WHO) ...".  Lines 63 & 64, "...people who had the influenza vaccine ...".  Line 67, "According to a study in Saudi Arabia ...".  Lines 74 & 75, "...ultimately, increases the trust in the ...".  Line 77, "...immunization during the flu season."  Lines 80 & 81, "...and July, 2021 in Aseer region, Saudi Arbia."  Line 93, "...should be included became 386."  Lines 94 & 95, "...number of participants was 422 subjects ...".  Line 103, "...study participants was 0.77."  Lines 117 & 118, "...as shown in Table 2.  Strongly agree ...".  Lines 131 & 132, "The participants were consented before the beginning ...".  Line 152, "...those who participate were Urbanites ...".  Line 154, "Furthermore, 21.5% had received flu vaccinations ...".  Lines 158 & 159, "...strongly agreed that the flu and COVID-19 vaccines ...".  Line 161, "Also, 54% strongly agreed that they ...".  Line 180, "...by social media when considering the decision ...".  Line 182, "participants with a lower education level, participants with a university ...".  Line 185, "...CI:0.158-0.693), respectively."  Line 186, "...participants with a postgraduate degree ...".  Line 187, "...attitude toward using social media as a source of information ...".  Line 202, "...those studies that has evaluated the ...".  Line 207, "...of health workers were willing to take ...".  Line 210, "...(48.8%) that the flu and COVID-19 ...".  Line 216, "...immunization during the flu season."  Line 219, "vaccine as per their history.  Previous ...".  Line 222, "...outcomes were reported in the USA, ...".  Lines 223 & 224, "...found statistically significance in age groups (36-45 years and > 55 years) as well as in the education level ...".  Line 227, "...most efficient way of preventing COVID-19 since ...".  Line 230, "...COVID-19 symptoms, with 75-81% coverage found for ...". Line 231, "...Pfizer and Moderna vaccines.  According to ...".  Line 232, "...days of hospitalization, ?% THERE IS NO % ENTERED HERE-PLEASE CLARIFY.  Line 239, "...found statistically significance in age groups ...".  Line 241, "... (Privet job) and in receiving the flu vaccine ...".  Line 245, "required to achieve herd immunity from the COVID-19 infection ...".  Line 253, "...at least one dose of the vaccine [37]."  Line 254, "...advised others to take a COVID-19 vaccine, whereas ...".  Line 260, "...COVID-19 cases during the flu season ...".  Lines 263 & 264, "According to a previous study ...".  Line 270, "...to the prevention of influenza.  The additional potential ...".  Line 272, "hesitant patients [41]."

Author Response

We would like to thank the reviewer for the
valuable comments. The manuscript went
through an extensive language editing and all
typos and comments were addressed.

Round 2

Reviewer 1 Report

The English spelling was improved. However, the significance of content and interest to the readers is similar. 

Regarding my previous comments, the year 2022 is about to end, and the present study raises awareness of the importance of vaccination to prevent both infections. But the data was collected two years ago. People's perceptions about COVID-19 have changed, as also the mortality. Please comment on this and how the study could still change predictions.

Regarding the results sections, the % sum between men and women has to be 100%. Same for the education level.

Author Response

Reply to reviewers

Manuscript ID: vaccines-2015832

Title:

Public Perception towards Vaccines as Preventive Measures against Twindemic (Seasonal Influenza and COVID-19): A Cross-Sectional Study from Aseer Region, Saudi Arabia

Reviewer 1:

Comment

Response

The English spelling was improved. However, the significance of content and interest to the readers is similar. 

We would like to thank the reviewer for his valuable comment. We think the manuscript’s idea and aims are significant. In addition, the manuscript has gone through an extensive editing and revision by an independent native English language speaker. If the reviewer has a specific part of the manuscript that he/she thinks it needs improvement, we will be more than welcoming to make it very clear.

Regarding my previous comments, the year 2022 is about to end, and the present study raises awareness of the importance of vaccination to prevent both infections. But the data was collected two years ago. People's perceptions about COVID-19 have changed, as also the mortality. Please comment on this and how the study could still change predictions.

We would like to thank the reviewer for the comment. We belief that the manuscript would help adding to the historical context of awareness and provide us with an indicator about how public perception toward vaccination has been changed. We aim for the futuristic study to explore the public perception toward vaccination with influence of different factors and study many predictors.

Regarding the results sections, the % sum between men and women has to be 100%. Same for the education level.

We would like to thank the reviewer for the comment. All percentages have been revised and corrected.